# Melatonin Inhibits Osteoclastogenesis and Osteolytic Bone Metastasis: Implications for Osteoporosis

**DOI:** 10.3390/ijms22179435

**Published:** 2021-08-30

**Authors:** Iona J. MacDonald, Hsiao-Chi Tsai, An-Chen Chang, Chien-Chung Huang, Shun-Fa Yang, Chih-Hsin Tang

**Affiliations:** 1Graduate Institute of Basic Medical Science, China Medical University, Taichung 40402, Taiwan; ionamac@gmail.com (I.J.M.); moxa0110@gmail.com (H.-C.T.); 2Translational Medicine Center, Shin Kong Wu Ho-Su Memorial Hospital, Taipei City 111, Taiwan; annone3212@gmail.com; 3School of Medicine, China Medical University, Taichung 40402, Taiwan; u104054003@cmu.edu.tw; 4Division of Immunology and Rheumatology, Department of Internal Medicine, China Medical University Hospital, Taichung 40447, Taiwan; 5Institute of Medicine, Chung Shan Medical University, Taichung 40201, Taiwan; ysf@csmu.edu.tw; 6Department of Medical Research, Chung Shan Medical University Hospital, Taichung 40201, Taiwan; 7Graduate Institute of Biomedical Sciences, China Medical University, Taichung 40402, Taiwan; 8Chinese Medicine Research Center, China Medical University, Taichung 40402, Taiwan; 9Department of Biotechnology, College of Health Science, Asia University, Taichung 41354, Taiwan

**Keywords:** apoptosis, bone mass protection, bone metastasis, immunomodulation, melatonin, osteoclastogenesis, osteoclasts, osteoporosis

## Abstract

Osteoblasts and osteoclasts are major cellular components in the bone microenvironment and they play a key role in the bone turnover cycle. Many risk factors interfere with this cycle and contribute to bone-wasting diseases that progressively destroy bone and markedly reduce quality of life. Melatonin (*N*-acetyl-5-methoxy-tryptamine) has demonstrated intriguing therapeutic potential in the bone microenvironment, with reported effects that include the regulation of bone metabolism, acceleration of osteoblastogenesis, inhibition of osteoclastogenesis and the induction of apoptosis in mature osteoclasts, as well as the suppression of osteolytic bone metastasis. This review aims to shed light on molecular and clinical evidence that points to possibilities of melatonin for the treatment of both osteoporosis and osteolytic bone metastasis. It appears that the therapeutic qualities of melatonin supplementation may enable existing antiresorptive osteoporotic drugs to treat osteolytic metastasis.

## 1. Introduction

The endogenous hormone melatonin (*N*-acetyl-5-methoxytryptamine) has long been recognized for its regulation of circadian and circannual functions [1]. Melatonin is mainly secreted by the pineal gland, but is also synthesized by extrapineal tissues and organs, including skin, the thymus, spleen, liver, bone marrow, and lymphocytes. As melatonin exhibits not only endocrine, but also autocrine or paracrine effects [2,3], it is considered to be an important regulator of the human immune system [3]. By activating the high-affinity G-protein-coupled melatonin MT_1/2_ receptors in target cells of the hypothalamic suprachiasmatic nucleus and the retina, melatonin regulates endocrine circadian rhythms and thus the sleep–wake cycle [4]. Activation of the MT_1/2_ receptors inhibits the adenylate cyclase (AC)/cyclic adenosine monophosphate (cAMP)/protein kinase A (PKA)/cAMP response element-binding protein (CREB) pathway and the guanylate cyclase (GC)/cyclic guanosine monophosphate (cGMP)/protein kinase G (PKG) pathway, while both receptors activate the phospholipase C (PLC) pathway, which increases production of the signaling molecules inositol trisphosphate (IP_3_) and diacylglycerol (DAG); IP_3_ triggers the release of intracellular calcium (Ca^2+^) via IP_3_ receptors, while DAG recruits and activates protein kinase C (PKC) [4,5]. Signaling cascade events that occur after the release of intracellular Ca^2+^ into the cytoplasm include the opening of receptor/ion channels and activation of the Ca^2+^/calmodulin-dependent kinase cascade [6]. PKC activation is necessary for angiotensin II-mediated, renal efferent arteriole vasoconstriction [7], phosphorylation of rat proximal tubule Na^+^-ATPase, which increases proximal tubule sodium reabsorption [8], and the modulation of aldosterone synthesis, with evidence controversially showing that PKC activation can both stimulate and inhibit aldosterone production [9]. Melatonin also affects nuclear signaling by binding to nuclear receptors, such as retinoid-related orphan receptor (ROR) receptors, which mediate metabolic processes, immunological functions and circadian outputs [4]. In particular, evidence suggests that RORα mediates the indirect (peripheral) effects of melatonin on immunomodulation, cellular proliferation, and bone differentiation, while membrane melatonin receptors appear to modulate the constitutive activity of RORα [10].

Melatonin supplementation has demonstrated therapeutic potential in the bone microenvironment [11], although this supporting evidence has been criticized for its absence, or because of its low to very low quality [12,13,14]. This review describes experimental and clinical evidence relating to melatonin-associated acceleration of osteoblastogenesis, inhibition of osteoclastogenesis and induction of apoptosis in osteoclasts, as well as the suppression of osteolytic bone metastasis. These features suggest further avenues for research into the potential use of this molecule in the treatment of osteoporosis.

## 2. Melatonin Metabolically Reprograms HSPC Self-Renewal, Stimulates Osteoblast Differentiation, and Inhibits Osteoclastogenesis

The onset of darkness metabolically reprograms the BM HSPCs to reacquire their undifferentiated state and repopulation potential, accompanied by limited proliferation, largely in response to melatonin-induced signaling [15,16]. Thus, daily light and dark cues and circadian rhythms regulate the differentiation and maintenance of BM-retained HSPCs [16]. It is also established that melatonin is locally produced by bone-forming stromal precursors in murine and human BM, and that melatonin preconditioning of mesenchymal cells increases their survival and therapeutic efficiency [16].

Nuclear factor kappa B (NF-κB) and the metabolic protein peroxisome proliferator-activated receptor gamma (PPARγ) appear to be the most important transcription factors involved in melatonin-induced stimulation of osteoblast differentiation. Preclinical evidence has shown that one way in which melatonin promotes osteoblast differentiation is by enhancing the expression of the osteogenic marker, Osterix, apparently through the protein kinase A (PKA) and PKC signaling pathways [17]. In ovariectomized (OVX) mice, melatonin reportedly suppressed osteoclastogenesis by inhibiting receptor activator of nuclear factor-kappa B ligand (RANKL)-induced tumor necrosis factor receptor-associated factor 6 (TRAF6), c-Jun N-terminal kinase (JNK), protein arginine methyltransferase 1 (PRMT1), and NF-κB signaling through a receptor-independent pathway [18]. In bone marrow monocytes (BMMs) isolated from the femurs and tibias of C57BL/6 mice, pharmacological concentrations (1 to 100 µM) of melatonin dose-dependently suppressed osteoclast differentiation and decreased numbers of tartrate-resistant acid phosphatase (TRAP)-positive cells as well as the gene expression of osteoclast-specific markers, via a reactive oxygen species (ROS)-mediated independent pathway [19]. Promisingly, melatonin suppressed estrogen deficiency-induced osteoporosis and promoted osteoblastogenesis in another study with OVX mice by suppressing the NLRP3 inflammasome in femoral bone protein through the regulation of Wnt/β-catenin signaling [20]. Other research has reported that melatonin enhanced osteogenic differentiation of human mesenchymal stem cells (MSCs) and restored oxidative stress-induced inhibition of osteogenesis by activating AMP-activated protein kinase (AMPK) and subsequently increasing forkhead box class O 3a (FOXO3a) and RUNX2 protein levels [21]. Melatonin-mediated enhancement of osteoblastogenesis has also been observed in human MSCs, in which melatonin can decrease PPARγ and increase RUNX expression, shifting MSCs towards osteogenesis [22]. Similarly, other investigations have recorded that melatonin inhibits adipogenic differentiation of human MSCs and enhances osteoblastogenesis by suppressing PPARγ expression [23,24]. However, contrasting findings have been reported by another research group using differentiating mouse embryo fibroblasts, in which high-dose melatonin (1 mM) treatment for 3 days significantly increased PPARγ expression and enhanced osteoblastogenesis [25]. This aspect of melatonin treatment needs further clarification.

It has been proposed that melatonin suppresses osteoclast differentiation by downregulating NF-κB and subsequently reducing the induction of nuclear factor of activated T cell cytoplasmic 1 (NFATc1), a transcription factor that is required for osteoclastogenesis [26]. Moreover, it appears that the anti-osteoclastogenic effect of melatonin occurs independently of its receptors MT_1_ and MT_2_, as silencing of these receptors in mouse bone marrow-derived macrophages failed to reverse the anti-osteoclastogenic signals of melatonin [26]. Other in vitro evidence suggests that melatonin inhibits RANKL-induced osteoclastogenesis by increasing the expression of Rev-erbα (a key circadian clock repressor) and reducing microRNA (miR)-882 expression in Raw264.7 cells [27]. Intriguingly, in vitro research has reported that melatonin inhibits osteoclastic activation under microgravity conditions by upregulating calcitonin and downregulating RANKL in osteoblasts, which indicates that melatonin may prevent bone loss during space flights [28].

The effects of melatonin on osteoblasts and osteoclasts may help to prevent and treat bone loss [22]. Melatonin is capable of regulating bone density by reducing oxidative stress in osteoclasts, promoting osteoclast cell differentiation and activity, and by increasing osteoblast-induced osteoprotegerin expression, preventing osteoclast precursors from differentiating into osteoclasts and inhibiting the process of bone resorption [22]. This apparently protective mechanism of melatonin against bone resorption suggests important therapeutic potential in bone-wasting diseases such as osteoporosis [22] and osteolysis [11,29]. Notably, by regulating inflammatory pathways and circadian rhythms, melatonin can stimulate the regeneration of cartilage and inhibit the release of proinflammatory cytokines or osteolytic factors including interleukin (IL)-1β, IL-8, tumor necrosis factor alpha (TNF-α), COX-2, matrix metalloproteinases (MMPs), and RANKL in joints by modulating the expression of key circadian *clock* genes, including *BMAL*, *CRY,* and/or *DEC2* [30,31]. Figure 1 illustrates the therapeutic effects of melatonin in the bone microenvironment. An important aspect to be considered for the potential therapeutic use of melatonin is its ability to promote apoptosis in mature osteoclasts and thus significantly reduce their lifespans, as well as suppress osteoclastic bone resorption [11]. This is discussed in the following section.

## 3. Melatonin Shows Potential for Osteoporosis—Preclinical and Clinical Evidence

Osteoporosis is a progressive bone disease, in which low bone mass and structural deterioration of bone tissue increases bone fragility and subsequent risk of fracture [32]. Although antiosteoporotic drugs (including bisphosphonates, selective estrogen receptor modulators, and calcitonin) inhibit bone resorption and assist with post-fracture bone healing [33,34,35], these medications fail to assist with bone formation [35] and they are associated with varying risks of adverse events, some of which are significant (e.g., osteonecrosis of the jaw, atypical fracture, and venous thromboembolic events) that greatly increase the risk of discontinuation [36]. New treatment strategies for osteoporosis are therefore needed that are as effective or more so, with the ability to promote bone formation without the side effects of existing medications.

Promisingly, the combination of melatonin with the bisphosphonate alendronate in OVX rats was associated with much less severe gastric damage than in OVX rats treated with alendronate alone [37]. Moreover, histological analyses of trabecular bone sections from the melatonin + alendronate group revealed similar bone matrix and architecture to that in the sham-operated controls, while the alendronate- or melatonin-treated rats had similar increases in trabecular thickness and reductions in apoptotic cells [37]. Thus, melatonin protected against OVX-induced gastric injury that was worsened by alendronate and melatonin provided similar supportive effects to those of alendronate concerning preservation of bone mass.

Melatonin has potential for the treatment of osteoporosis and may also help to prevent this disease. For instance, melatonin and calcium carbonate treatment of osteoporotic rats was associated with increased antioxidative stress activities, improved lumbar vertebrae and femur bone densities, and upregulated serum calcium and bone mineral levels, compared with osteoporotic rats administered melatonin or calcium carbonate alone [38]. Preclinical [27,39,40,41,42,43,44,45,46,47,48,49,50,51,52,53,54,55,56,57,58,59,60] and clinical evidence [61,62,63,64] indicate an antiosteoporotic role for melatonin (see Table 1). Melatonin modified bone remodeling induced by ovariectomy in rats, in which a pharmacological dose of melatonin (25 µg/mL drinking water) given for up to 60 days after surgery prevented postsurgical increases in urinary deoxypyridinoline (a marker of bone resorption) and increased serum phosphorus and bone alkaline phosphatase (BAP) levels compared with untreated rats [39]. In further experiments, by the same researchers, where OVX rats were administered melatonin (25 µg/mL drinking water) with or without estradiol (10 µg/kg subcutaneously (s.c.); 5 days/week) for up to 60 days, melatonin potentiated estradiol-induced inhibition of OVX-induced bone resorption and impaired estradiol-induced increases in serum phosphorus [40]. Both melatonin and estradiol lowered serum BAP activity; melatonin also augmented spinal bone area values and bone mineral content of the whole of the skeleton and tibia, with the highest values seen in rats given both melatonin and estradiol [40]. In rats treated with methylprednisolone (5 mg/kg s.c., 5 days/week) for 10 weeks, the addition of melatonin (25 µg/mL drinking water) augmented bone protective effects and lowered still further the circulating levels of C-telopeptide fragments of collagen type I (CTX, an index of bone resorption) observed with methylprednisolone alone [41]. In a mouse model of OVX-induced osteoporosis, orally administered melatonin completely reversed the architectural deterioration and functional defects in bone by specifically increasing bone formation [44]. As the study researchers suggest, their finding that bone mass is regulated via the pineal-derived melatonin-MT_2_ receptor pathway needs further exploration to clarify whether more selective receptor agonist therapy or oral melatonin treatment activates MT_2_ and can treat postmenopausal osteoporosis [44]. Melatonin may also be appropriate in the treatment of type 2 diabetic osteoporosis, with evidence of low to high doses of melatonin improving bone microstructure and promoting bone formation in the osteoblastic cell line MC3T3-E1 (at concentrations of 1, 10, and 100 µM) exposed for 48 h to high glucose (25.5 mM) and in a diabetic rat model (10 and 50 mg/kg) [56]. The study findings identified that high glucose induces ferroptosis (an iron- and ROS-dependent form of regulated cell death) by increasing ROS/lipid peroxidation/glutathione depletion in type 2 diabetic osteoporosis [56]. Importantly, melatonin inhibited ferroptosis of osteoblasts and improved the osteogenic capacity of MC3T3-E1 cells by activating nuclear factor erythroid 2-related factor 2 (Nrf2) and heme oxygenase-1 (HO-1) signaling in vitro and in vivo [56]. Melatonin also protects MC3T3-E1 cells against high glucose-induced changes (reduced viability, apoptosis and calcium influx) by inhibiting protein kinase RNA-like endoplasmic reticulum kinase (PERK)-eukaryotic initiation factor 2 alpha (eIF2α)-activating transcription factor 4 (ATF4)-C/EBP homologous protein (CHOP) signaling, a major endoplasmic reticulum stress pathway that is needed for cell survival [57]. Melatonin may also be helpful for treating glucocorticoid-induced osteoporosis, as melatonin treatment reportedly rescued MC3T3-E1 cells from dexamethasone-induced inhibition of osteoblast differentiation and mineralization via the phosphatidylinositol-3-kinase (PI3K)/protein kinase B (AKT) and bone morphogenetic protein (BMP)/Smad signaling pathways [58].

Further evidence suggesting that melatonin has the potential to promote bone regeneration comes from studies in mice, in which intraperitoneal (i.p.) melatonin increased the formation of mouse cortical bone formation in vivo [42]. Promisingly, melatonin treatment increases bone mass around implants. In adult Beagle dogs, topical application of lyophilized powdered melatonin (1.2 mg) and recombination human growth hormone (4 IU) to osteotomy sites before dental implants synergistically enhanced new bone formation around the implants within 5 weeks of insertion [43], while in OVX rats, melatonin (50 mg/kg, i.p.) increased the generation of new bone around prostheses, enhanced osteoblast proliferation and increased implant fixation strength [48]. Interestingly, a composite adhesive hydrogel system (GelMA-DOPA@MT) that releases melatonin in a sustained fashion in a local area has shown that it reduces apoptosis in osteoblasts and increases bone mass around the implant in OVX rats treated with this composite system [51], which suggests that it may correct implant loosening in patients with osteoporosis. Bone loss (aseptic loosening or periprosthetic osteolysis) is a significant complication following total joint replacement and a major cause of implant failure. Importantly, wear particles stimulate bone marrow mesenchymal stem cells (BMMSCs) in the bone-prosthesis interface and impair their osteogenic potential [52]. Interestingly, melatonin can reportedly improve the matrix mineralization and expression of osteogenic markers in BMMSCs exposed to titanium (Ti) wear particles in vitro, and ameliorate Ti particle-induced osteolysis in a murine calvarial osteolysis model [52]. Melatonin-treated BMMSCs exhibited increases in levels of silent information regulator type 1 (SIRT1) and intracellular antioxidant enzymes, particularly superoxide dismutase 2 (SOD2), highlighting the importance of the SIRT1/SOD2 signaling pathway in bone mass and wear particle-induced osteolysis around prostheses [52]. Currently, no therapeutic agents exist that are capable of protecting against inflammatory bone loss diseases. Melatonin has shown potential in this scenario. In a series of experiments involving osteogenic precursor cells (human bone MSCs and MC3T3-E1 preosteoblasts), melatonin promoted osteogenic differentiation and mineralization in inflammatory conditions; a process in which Wnt4 was essential, via activation of β-catenin and p38-c-Jun N-terminal kinase (JNK) MAPK signaling [65]. Moreover, melatonin has been found to effectively reduce accumulation of reactive oxygen species (ROS) during osteogenesis of BMMSCs in the presence of TNF-α, by upregulating antioxidase expression and downregulating oxidase expression [55].

Melatonin may protect against natural age-related osteoporosis, according to a study that used naturally aged male mice [66]. Orally administered melatonin in drinking water from 4 to 20 months of age improved bone strength and trabecular bone density of the femur, apparently via the MT_2_ receptor, which was detected in osteoblasts and osteoclasts in the femur bones [66]. In aged rats, micro-CT data have recorded increases in BMD, bone volume/tissue volume (BV/TV), trabecular number (Tb.N) and trabecular thickness (Tb.Th), as well as reductions in the Structure Model Index (SMI) and trabecular separation/spacing (Tb.Sp) values after melatonin administration [45]. That study also reported that melatonin was associated with reductions in calcium and phosphorus losses in urine, increases in serum BAP and osteocalcin (OCN) levels and increases in bone formation and bone mineralization rates [45]. Melatonin also upregulated osteogenic differentiation gene expression and downregulated adipogenic differentiation gene expression [45]. In another study, micro-CT scans of OVX mice found that compared with high-dose melatonin (100 mg/kg/day), low-dose melatonin (10 mg/kg/day) produced more marked increases in BMD, higher BV/TV and Tb.N values, and greater reductions in Tb.Sp values [53]. Serum levels of the bone formation marker N-terminal propeptide of type I procollagen (PINP) were consistent with micro-CT data [53]. Thus, even at a low dose, melatonin appears to be effective against osteoporosis. Some preclinical evidence suggests that the addition of rapamycin enhances the anti-osteoporotic effects of melatonin in age-dependent osteoporosis [67]. In OVX rats, melatonin combined with rapamycin was more effective than melatonin alone in improving osteoporotic bone microarchitecture and bone quality and strength [67]. The beneficial effects of melatonin on age-dependent bone loss were achieved through regulation of the osteoprotegerin (OPG)/RANKL signaling pathway [67].

In mice with retinoic acid-induced osteoporosis, melatonin prevented bone destruction by suppressing levels of bone loss, repairing the trabecular microstructure, and promoting bone formation, all of which the study researchers speculated involves extracellular signal-regulated kinase (ERK)/Smad activation and NF-κB signaling [47]. Melatonin has indicated promise as a treatment for postmenopausal osteoporosis, by preserving antioxidant function and osteogenic potential of BMMSCs from OVX rats, in which intravenous (i.v.) injections of melatonin via the tail vein ameliorated bone microarchitecture in the femur [60]. Interestingly, melatonin apparently promotes the osteogenic differentiation of BMMSCs by upregulating microRNA-92b-5p expression, which then enhances the differentiation of BMMSCs into mature osteoblasts by targeting intracellular adhesion molecule-1 (ICAM-1) [54]. In another study, melatonin promoted the osteogenic differentiation of BMMSCs and prevented the progression of osteoporosis by significantly decreasing the expression of circ_0003865, which subsequently increased levels of osteogenic marker genes (*ALP*, *RUNX2* and *OPN*) and marked induction of osteogenic differentiation [59]. Moreover, melatonin-induced silencing of circ_0003865 increased the expression of miR-3653-3p, which substantially increased *ALP*, *RUNX2*, and *OPN* expression and promoted BMMSC osteogenic differentiation [59].

Results of a mouse study suggest that restoring melatonin levels in melatonin-deficient mice prevents scoliosis and improves bone density [46]. Similarly, serum melatonin levels are low in patients with multiple sclerosis (MS) and are not only associated with disease severity, but are also inversely correlated with serum procalcitonin levels (procalcitonin is involved in calcium homeostasis) [68]. Patients with osteoporosis and chronic renal failure on maintenance hemodialysis have been found to have lower serum melatonin levels and higher serum levels of advanced oxidation protein products, malondialdehyde, and proinflammatory cytokines (IL-1, IL-6, and TNF-α) compared with their non-osteoporotic counterparts [69]. In a mouse model of MS induced by experimental autoimmune encephalomyelitis (EAE), melatonin treatment (10 mg/kg/day, i.p.) normalized bone metabolites and reduced the risk of osteoporosis by increasing serum levels of calcium, 25-hydroxyvitamin D and OCN, and melatonin treatment reduced serum procalcitonin levels [68]. Another mouse study found that melatonin protects against postmenopausal bone loss by increasing osteoblast citrate production and enhancing the closely related process of matrix mineralization [50]. The study researchers suggested that as citrate secretion in osteoblasts increases via the upregulation of zinc transporter (ZIP-1) with subsequent accumulation of intracellular zinc, it is worth investigating the possibility of promoting citrate secretion in osteoblasts through dietary zinc supplementation in osteoporosis treatment [50].

Clinical trials have pointed to the benefits of oral melatonin supplementation on bone, reflected by reduced bone turnover and increases in serum bone formation markers, increases in BMD, and a fall in the long-term likelihood of vertebral fracture risk [61,62,64]. However, some controversy surrounds the clinical trial evidence as to the efficacy and safety of melatonin in humans. First of all, the scarcity of double-blind, randomized, placebo-controlled trials makes it difficult to draw firm conclusions as to the effects of melatonin supplementation, especially since the lack of numbers affects the statistical power [4]. In regard to safety, trials have been criticized for their weak methodology around the reporting of adverse events and for not including a priori consideration of which adverse events should be determined as relevant [4]. Moreover, just one clinical trial has considered the safety of melatonin as the primary outcome; in that trial, orally administered melatonin at 10 mg/day for 4 weeks in healthy volunteers did not produce any adverse outcomes that might compromise its daily use at this dose over a 4-week period [13]. Reviews of the clinical evidence generally agree that short-, intermediate-, and long-term oral melatonin supplementation (for days, weeks-to-months, and years, respectively) is safe and associated with only mild, transient adverse effects such as dizziness, sleepiness, nausea, and headache; no dose has been linked to serious adverse effects [12,14]. Nonetheless, melatonin has been linked to endocrine disturbances including reproductive parameters and glucose metabolism, as well as cardiovascular dysfunction (variability in heart rate and blood pressure), apparently in response to dosage, timing of dosing and possibly melatonin-antihypertensive drug interactions [12]. Following dosing schedules that imitate normal circadian rhythms is advisable for avoiding or managing most of the adverse effects associated with melatonin [12]. As discussed in the following section, disruption of circadian rhythms is critical for bone oncogenesis and metastatic disease.

## 4. Melatonin Shows Potential in Bone Cancer Treatment and Osteolytic Bone Metastasis

Disrupting circadian rhythms with exposure to artificial light at night (LAN) can promote the risk of cancer growth and progression, largely through the alteration of nocturnal melatonin biosynthesis [70]. Tibias of *Foxn1^nu^* athymic nude female mice (which produce marked night-time circadian melatonin signals) were injected with estrogen receptor-positive human breast cancer cells (to mimic bone metastatic disease) and the mice were housed in bright light for 12 h each day followed by either 12 h of complete darkness or 12 h of dim artificial light (LAN, 0.2 lux) [71,72]. Whereas, the mice assigned to 12 h of darkness produced high levels of endogenous nocturnal melatonin, this was suppressed in the mice assigned to 12 h of dim light, which developed highly osteolytic, fully developed breast cancer tumors in bone according to measurements of tumor bioluminescence using the in vivo imaging system (IVIS), X-rays, and micro-computed tomography (CT) images [71,72]. Moreover, the tumors in the mice experiencing 12 h of dim light at night were treatment-resistant to doxorubicin, but when doxorubicin was delivered to these mice in a chronotherapeutic schedule in circadian alignment with nocturnal melatonin, bone tumors grew more slowly, bone damage was decreased and new bone formation was observed [71]. Blocking the MT_1/2_ receptors with the melatonin receptor antagonist luzindole prevented nocturnal melatonin from effectively inhibiting metastatic tumor growth in the bone [72]. Other evidence supports the contention that artificial light at night drives intrinsic resistance to chemotherapy including doxorubicin, paclitaxel, and tamoxifen [73,74,75]. Clearly, disrupting circadian rhythms deregulates bone homeostasis and influences oncogenesis.

Melatonin has shown marked in vitro and in vivo activity against osteosarcoma (one of the most common primary malignant bone tumors), and is considered to have promising potential as an adjuvant with conventional chemotherapy in anti-osteosarcoma regimens [76]. The high propensity for osteosarcoma to metastasize, particularly to the lung, is linked to treatment failure and high mortality [76,77]. The prognosis is very poor for patients with osteosarcoma and pulmonary metastases, even after treatment by metastasectomy and chemotherapy [78], with more than half of these cases relapsing [77]. Thus, current best treatment strategies for osteosarcoma need to be greatly improved. A greater understanding of the intracellular pathways related to the metastatic transfer of osteosarcoma cells has focused attention on melatonin as a novel, non-toxic addition to conventional chemotherapy for osteosarcoma, capable of direct oncostatic effects on the neoplastic cell and indirect tumoricidal effects via immunostimulation [79], as well as augmentation of anti-osteosarcoma cancer agents and amelioration of chemotherapy-related side effects [76]. As shown in Table 2, melatonin suppresses osteosarcoma cells by modulating various signaling pathways and mechanisms. In human osteosarcoma Saos-2 cells, melatonin dose-dependently inhibits cellular activity, exerts cytostasis by increasing cell accumulation in the G1 phase but decreasing those in the S phase, and induces apoptosis [80]. Melatonin also appears to be promising for metastatic osteosarcoma. For instance, melatonin inhibited osteosarcoma metastasis in a mouse model of osteosarcoma by downregulating SOX9-mediated signaling [60] and, in another study, melatonin inhibited the migratory potential and invasiveness of human osteosarcoma HOS and U2OS cells, and suppressed C-C motif chemokine ligand 24 (CCL24) levels in U2OS cells by inhibiting the JNK pathway, preventing osteosarcoma invasion [81]. Melatonin has also shown promise in another common primary malignant bone tumor, Ewing’s sarcoma, inducing apoptosis in Ewing’s sarcoma cells via upregulation of the death receptor Fas and its ligand Fas L [82].

Novel treatments are needed that can target the intracellular signaling pathways regulating the invasion-metastasis cascade. In relation to this, melatonin has shown promising therapeutic potential by decreasing osteoclast differentiation, bone resorption activity and promoting apoptosis in mature osteoclasts [11]. Moreover, melatonin can downregulate the p38 mitogen-activated protein kinase (MAPK) pathway and thus inhibit RANKL production in lung and prostate cancer cells, preventing cancer-associated osteoclast differentiation [11]. Promisingly, mouse models of lung and prostate bone metastases administered twice-weekly melatonin displayed marked reductions in tumor volumes and numbers of osteolytic lesions, as well as numbers of TRAP-positive osteoclasts in tibia bone marrow and tumor tissue RANKL expression [11]. Table 2 lists cancer-secreted factors that are inhibited by melatonin. Although these factors are associated with osteolytic metastasis, it remains to be confirmed as to whether melatonin-induced inhibition of these factors is associated with osteolytic metastasis.

## 5. Conclusions

Emerging research indicates that melatonin inhibits the expression of osteolytic factors produced by tumor cells and suppresses tumor growth, pointing to the potential of melatonin for the treatment of osteolytic metastasis. In clinical applications, melatonin treatment of postmenopausal women with osteopenia indicates improvements in bone health, which can improve osteoporosis. We predict from the molecular evidence that this endogenous hormone has tremendous potential, by enabling existing antiresorptive osteoporotic drugs to effectively treat osteolytic metastasis. This aspect remains to be explored in future preclinical work.

## Figures and Tables

**Figure 1 ijms-22-09435-f001:**
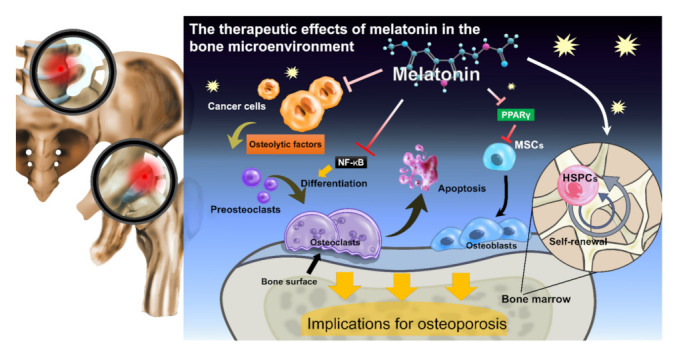
Melatonin suppresses proinflammatory cytokine production, tumor-secreted osteolytic factor expression, and bone metastatic tumor growth.

**Table 1 ijms-22-09435-t001:** Summary of preclinical and clinical evidence demonstrating bone-protective effects of melatonin.

**Preclinical**					
**Cellular or Animal Model**	**Dosage**	**Administration**	**Outcomes**		**Ref.**
BMMSCs	10 µmol/L	Culture	Melatonin promoted osteogenesis in BMMSCs by upregulating miR-92b-5p expression, which enhanced the differentiation of BMMSCs into mature osteoblasts by targeting ICAM-1.		[54]
MC3T3-E1 cells, human bone MSCs	1 mM	Culture	Melatonin promoted osteogenic differentiation and mineralization in inflammatory conditions; this process required Wnt4, via activation of β-catenin and p38-JNK MAPK signaling.		[65]
BMMSCs	100 μmol/L	Culture	Melatonin inhibited ROS generation during osteogenesis of BMMSCs in the presence of TNF-α and promoted the osteogenic differentiation of BMMSCs.		[55]
MC3T3-E1 cells Diabetic rats	Cells: 1, 10, or 100 μM Rats: 10 or 50 mg/kg	Culture Animals: Intraperitoneal injection	Melatonin inhibited ferroptosis and improved the osteogenic capacity of MC3T3-E1 cells by activating Nrf2/HO-1 signaling in vitro and in vivo.		[56]
MC3T3-E1 cells	100 nM	Culture	Melatonin protects MC3T3-E1 cells against high glucose-induced changes (reduced viability, apoptosis and calcium influx) by inhibiting the PERK-eIF2α-ATF4-CHOP signaling pathway.		[57]
MC3T3-E1 cells	1 μM	Culture	Melatonin treatment rescued MC3T3-E1 cells from dexamethasone-induced inhibition of osteoblast differentiation via the PI3K/AKT and BMP/Smad signaling pathways.		[58]
OVX rats	25 μg/mL	Oral (melatonin in drinking water)	Melatonin and alendronate provided similar supportive effects on preservation of bone mass in OVX rats, with no additive effect on bone remodeling when these treatments were combined. However, melatonin prevented alendronate-induced gastric side effects.		[37]
Raw264.7 cells	0.1 or 1 µmol	Culture	Melatonin inhibits RANKL-induced osteoclastogenesis by increasing the expression of Rev-erbα and reducing miR-882 expression in Raw264.7 cells.		[27]
BMMSCs	100 μmol/L	Culture	Melatonin promoted BMMSC osteogenic differentiation and inhibited osteoporosis pathogenesis by suppressing the expression of circ_0003865 and increasing the expression of miR-3653-3p.		[59]
**Clinical**					
**Study Population**	**Route of Administration**	**Dosing Schedule**	**Outcomes**	**ClinicalTrials.gov. Identifier**	**Ref.**
Perimenopausal women (*n* = 18)	Oral	Nightly placebo or melatonin (3 mg) for 6 months.	At 6 months, serum markers of bone resorption (NTX) and formation (OCN) were not significantly changed from baseline in either group, although the NTX:OCN ratio trended downward over time with melatonin (placebo showed no such trend). Melatonin also reduced osteoclast to osteoblast ratios.	NCT01152580	[61]
Postmenopausal women with osteopenia* (*n* = 81)	Oral	Nightly placebo or melatonin (1 mg or 3 mg) for 12 months. All study participants also received a daily supplement of 800 mg calcium and 20 μg vitamin D_3_.	After 1 year of treatment, femoral neck BMD was increased by 0.5% with 1 mg/day melatonin and by 2.3% with 3 mg/day melatonin, compared with placebo. At 12 months, trabecular thickness was increased from baseline in the (combined) melatonin group by 2.2% compared with placebo, while volumetric BMD at the lumbar spine was increased by 3.6% in the 3 mg/day melatonin group compared with placebo. Biochemical markers of bone turnover were not affected by melatonin, although 24-h urinary calcium was decreased by 3.7% in the (combined) melatonin group compared with placebo.	NCT01690000	[62]
Postmenopausal women with osteopenia * (*n* = 23)	Oral	Nightly placebo or MSDK supplementation: melatonin (5 mg), strontium (citrate), vitamin D_3_ and vitamin K_2_.	Over 1 year, compared with placebo, MSDK treatment increased lumbar spine BMD by 4.3% and left femoral neck BMD by 2.2%, and showed a trend towards an increase in hip BMD from baseline. The 10-year vertebral fracture risk probability fell by 6.48% with MSDK treatment, but increased by 10.8% with placebo. MSDK increased serum bone formation markers and reduced bone turnover.	NCT01870115	[64]

* Osteopenia was defined as low bone mass with a T-score between −1 and −2.5 in either the hip or spine. BMMSCs, bone marrow mesenchymal stem cells; ICAM-1, intracellular adhesion molecule-1; JNK, c-Jun N-terminal kinase; MAPK, mitogen-activated protein kinase; ROS, reactive oxygen species; TNF-α, tumor necrosis factor alpha; Nrf2, nuclear factor erythroid 2-related factor 2; HO-1, heme oxygenase-1; PERK, protein kinase RNA-like endoplasmic reticulum kinase; eIF2α; eukaryotic initiation factor 2 alpha; ATF4, activating transcription factor 4; CHOP, C/EBP homologous protein; PI3K, phosphatidylinositol-3-kinase; AKT, protein kinase B; BMP, bone morphogenetic protein; OVX, ovariectomized; RANKL, receptor activator of nuclear factor-kappa B ligand; NTX, N-terminal telopeptide; OCN, osteocalcin; BMD, bone mineral density.

**Table 2 ijms-22-09435-t002:** Melatonin treatment reduces cancer-associated osteolytic factors.

Type of Cancer	Osteolytic Factors	Dosage	Model	Outcomes	Ref.
Pancreatic, cervical, lung	VEGF HIF-1α	Cells: 1 nM or 1 mM	Cell lines: PANC-1, HeLa and A549	At the high concentration (1 mM), melatonin inhibited VEGF mRNA and protein levels, as well as HIF-1α protein, in all three human cancer cell lines.	[83]
Prostate	HIF-1α	Cells: 1 mM	Cell lines: DU145, PC-3, and LNCaP	Melatonin-induced inhibition of HIF-1α protein expression, HIF-1α transcriptional activity and the release of VEGF in all three cell lines correlated with dephosphorylation of p70S6K and its direct target RPS6.	[84]
Bladder	COX-2	Cells: 1 mM	Cell lines: T24, UMUC3 and 5637	When combined with curcumin, melatonin enhanced the inhibitory effects of curcumin on COX-2 activity and enhanced the antiproliferative, antimigratory and proapoptotic activities of curcumin in bladder cancer cells.	[85]
		Animal: 10 mg/kg	Animal: BALB/c nude mice		
Osteosarcoma	SOX9	Cells: 0.5 mM	Cell lines: HOS and U2-OS	Melatonin suppressed osteosarcoma cell migration and invasion and also significantly inhibited osteosarcoma metastasis in a mouse model of osteosarcoma. These effects were achieved by downregulating SOX9-mediated signaling.	[86]
		Animal: 100 mg/kg	Animal: BALB/c nude mice		
Gastric adenocarcinoma	MMP-2 MMP-9	Cells: 0.1, 0.5 or 1.5 mM	Cell lines: MGC80-3 and SGC-7901	Melatonin suppressed IL-1β-induced EMT in human gastric adenocarcinoma cells by targeting IL-1β/NF-κB/MMP-2/MMP-9 signaling.	[87]
Osteosarcoma	CCL24	Cells: 2 mM	Cell lines: HOS and U2OS	Melatonin inhibited the migratory potential and invasiveness of osteosarcoma HOS and U2OS cells. Melatonin also suppressed chemokine CCL24 levels in U2OS cells through the inhibition of the JNK pathway.	[81]
Oral	MMP-9	Cells: 100 and 250 μg/mL	Cell line: SAS	Areca nut extract components (betel quid chewing) may contribute to tumor invasion and metastasis by stimulating MMP-9 mRNA expression and secretion of oral cancer cells, which was inhibited by melatonin.	[88]
Osteosarcoma	MMP-9 HIF-1αTGF-β	Cells: 50, 100, 200, 500 and 1000 nM	Cell line: MG-63	Melatonin inhibits TGF-β1-induced EMT in osteosarcoma MG-63 cells by suppressing HIF-1α/Snail/MMP-9 signaling.	[49]
Prostate	MMP-13	Cells: 1 mM	Cell lines: DU145 and PC-3	Melatonin inhibited the migratory and invasive properties of prostate cancer cells, as well as MMP-13 expression, via the MT_1_ receptor and PLC, p38, and c-Jun signaling. Melatonin also inhibited prostate cancer metastasis and MMP-13 expression in an orthotopic prostate cancer model.	[89]
		Animal: 20 or 60 mg/kg	Animal: SCID mice		
Breast	IL-6	Animal: 5 mg/kg	Animal: Female rats with DMBA-induced breast cancer	Combined zinc and melatonin therapy helped to prevent tumor growth by significantly disrupting the metabolism of several elements (iron, magnesium, zinc and copper), and by suppressing IL-6 levels and reducing tissue damage that encourages tumor growth.	[90]
Lung, prostate	RANKL	Cells: 0.1, 0.3 or 0.7 mM	Cell lines: A549 and PC-3	Melatonin inhibited RANKL production in lung and prostate cancer cells by downregulating the p38 MAPK pathway, which consequently prevented cancer-associated osteoclast differentiation. In animal models of lung and prostate bone metastasis, melatonin treatment markedly reduced tumor volumes and numbers of osteolytic lesions.	[11]
		Animal: 20 or 60 mg/kg	Animal: BALB/c nude mice		
Breast	Integrin β 1Elf-5	Cells: 5 mM	Cell lines: MCF-7 and MDA-MB-231	MEMP HT (5 mg melatonin, 0.5 mg estradiol, and 50 mg progesterone [half the recommended dose] hormone therapy) showed anticancer activity in ER^+^ and triple negative breast cancer cells. These effects were largely attributed to the melatonin component and MEMP HT working through MEK1/2- and MEK-5-dependent intracellular signaling cascades in each cancer cell line, modulating intracellular signaling proteins that encourage the inhibition of cellular proliferative and migratory activities.	[91]
Pancreatic stellate cells	COX-2 IL-6 TNF-α	Cells: 1000, 100, 10 or 1 μM	Cells: Primary PSCs from Wistar rat pups (3–5 days after birth)	Pharmacological concentrations of melatonin increased ROS production and reduced levels of glutathione in PSCs under hypoxic conditions. Melatonin downregulated NF-kB phosphorylation and COX-2, IL-6, and TNF-α expression.	[92]
Gastric	TGF-β1	Cells: 2 or 4 mM	Cell line: MFC	Melatonin inhibited gastric cancer cell proliferation in vitro by increasing TGF-β1 expression and also increased TGF-β1 levels in gastric cancer tumor tissues in vivo.	[93]
		Animal: 25, 50, or 100 mg/kg	Animal: H-2K^k^ mice		

VEGF, vascular endothelial growth factor; HIF-1α, hypoxia-inducible factor 1-alpha; mRNA, messenger RNA; COX-2, cyclooxygenase-2; MMP, matrix metalloproteinase; IL-1β, interleukin 1 beta; EMT, epithelial-to-mesenchymal transition; NF-κB, nuclear factor kappa B; CCL24, C-C motif chemokine ligand 24; JNK, c-Jun N-terminal kinase; TGF-β1, transforming growth factor beta-1; MT_1_ receptor, high-affinity G-protein-coupled melatonin receptor; PLC, phospholipase C; DMBA, 7,12-dimethylbenz(a)anthracene; RANKL, receptor activator of nuclear factor-kappa B ligand; MAPK, mitogen-activated protein kinase; ER^+^, estrogen receptor-positive; PSCs, pancreatic stellate cells; ROS, reactive oxygen species.

## Data Availability

Not applicable.

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
