# Peer review of "Melatonin Inhibits Osteoclastogenesis and Osteolytic Bone Metastasis: Implications for Osteoporosis"

_ijms, 2021, doi:10.3390/ijms22179435_

Round 1
Reviewer 1 Report
This review is on an intriguing topic which deserves consideration. However, the current text has several shortcomings which limit the value of the report.
1. The discussion is often “off topic’ and touches on a large number of aspects of melatonin’s properties. Effects on mood, obesity, cardiovascular disease, sleep cycles, and cancer are not pertinent to the issue of osteoporosis and distract from the intended focus of the report. The irrelevant sections should be omitted so the target of the review can be made more clear.
2. A review should be analytical and not merely list relevant papers. For example, the text of the paper by Girardo, Bettini, Dema and Cervellati , (2011) concludes “No permanent deficiency of secretion of melatonin occurs in patients with AIS.” Yet the current review citing this report as evidence, states: ”Melatonin deficiency is implicated in the development of adolescent idiopathic scoliosis”.
3. A critical look at some of the papers cited raises questions. For example, the paper of Amstrup, Sikjaer, Heickendorff, Mosekilde and Rejnmark, 2015 states: “Compared to placebo, femoral neck BMD increased by 1.4% in response to melatonin (P < 0.05). With a total study population of only 81, women, the reported significance of 1% seems unlikely to have any meaning at all. There are several other examples of questionable reports. These studies need to be carefully read not merely cited. Otherwise defective papers live on indefinitely.
4. The document periodically segues into discussion osteolytic bone metastasis which muddles the discussion of osteoarthritis. These areas are poorly integrated in the current report. Various tumors can lead to both osteolytic or osteoblastic bone disease.
5. There have been several recent reviews that cover the same ground as the current submission. E.g. Lu X, Yu S, Chen G, Zheng W, Peng J, Huang X, Chen L. Insight into the roles of melatonin in bone tissue and bone related diseases (Review). Int J Mol Med. 2021;47:82. doi: 10.3892/ijmm.2021.4915. Therefore, it is important to emphasize what is original and novel in this submission.
Overall, the report needs to be tightened and directed to the main topic. There should be evidence that the authors have carefully studied cited papers. If there were fewer and more carefully selected references, these could then be more searchingly discussed.
Author Response
This review is on an intriguing topic which deserves consideration. However, the current text has several shortcomings which limit the value of the report.
- The discussion is often “off topic’ and touches on a large number of aspects of melatonin’s properties. Effects on mood, obesity, cardiovascular disease, sleep cycles, and cancer are not pertinent to the issue of osteoporosis and distract from the intended focus of the report. The irrelevant sections should be omitted so the target of the review can be made more clear.
Ans: We thank the Reviewer for this observation and have accordingly removed those sections of the text that were irrelevant regarding the implications of melatonin for osteoporosis. Thus, the Introduction has been considerably shortened and we have removed mention of melatonin-induced effects in the other sections, wherever those effects do not directly implicate the issue of osteoporosis. All amendments are marked-up in blue font.
- A review should be analytical and not merely list relevant papers. For example, the text of the paper by Girardo, Bettini, Dema and Cervellati , (2011) concludes “No permanent deficiency of secretion of melatonin occurs in patients with AIS.” Yet the current review citing this report as evidence, states: ”Melatonin deficiency is implicated in the development of adolescent idiopathic scoliosis”.
Ans: We thank the Reviewer for this thoughtful consideration and we have accordingly deleted the statement under question (with its associated reference).
- A critical look at some of the papers cited raises questions. For example, the paper of Amstrup, Sikjaer, Heickendorff, Mosekilde and Rejnmark, 2015 states: “Compared to placebo, femoral neck BMD increased by 1.4% in response to melatonin (P < 0.05). With a total study population of only 81, women, the reported significance of 1% seems unlikely to have any meaning at all. There are several other examples of questionable reports. These studies need to be carefully read not merely cited. Otherwise defective papers live on indefinitely.
Ans: We thank the Reviewer for highlighting this issue and we have accordingly ensured that where we cite and discuss results from papers that describe therapeutic qualities of melatonin, we have avoided describing those qualities as definitive. Instead, we have described them as being indicative of positive outcomes for bone health. We have also removed references to “significant” study results, in view of the fact that those studies are either cellular or preclinical experiments, or involve very small clinical populations.
- The document periodically segues into discussion osteolytic bone metastasis which muddles the discussion of osteoarthritis. These areas are poorly integrated in the current report. Various tumors can lead to both osteolytic or osteoblastic bone disease.
Ans: We thank the Reviewer for this thoughtful comment and we have made sure that there is no mention of osteoarthritis in our text. We have also revised the text to highlight the potential of melatonin treatment for both osteoporosis and osteolytic metastasis.
- There have been several recent reviews that cover the same ground as the current submission. E.g. Lu X, Yu S, Chen G, Zheng W, Peng J, Huang X, Chen L. Insight into the roles of melatonin in bone tissue and bone related diseases (Review). Int J Mol Med. 2021;47:82. doi: 10.3892/ijmm.2021.4915. Therefore, it is important to emphasize what is original and novel in this submission.
Ans: We thank the Reviewer for this insightful comment. The novel part of this review is the “Melatonin shows potential for osteoporosis – preclinical and clinical evidence” section. We have rephrased our Abstract and Conclusion to better reflect the novelty of this review in comparison with other recent reviews. As we indicate in our review, evidence from melatonin-associated inhibition of osteolytic bone metastasis may predict the potential of melatonin for osteoporosis management.
Overall, the report needs to be tightened and directed to the main topic. There should be evidence that the authors have carefully studied cited papers. If there were fewer and more carefully selected references, these could then be more searchingly discussed.
Ans: We thank the Reviewer for this insight and we have accordingly deleted the irrelevant references, which has considerably tightened the review and directed its discussion to the main topic.
Reviewer 2 Report
Dear Authors,
This is an interesting review describing the potential therapeutic effects of melatonin in bone associated diseases. In general the review was well-developed, however, I suggest the following changes:
- After talking about HSPC, I would include the osteoporosis section. The cancer section should be the last one.
- Respect to the cancer section, you mostly focused on metastasis. I would include more references related to melatonin and bone cancer treatment first (osteosarcoma, Ewing Sarcoma,...), and after this sub-section, developed the bone metastasis sub-section, as the origin of metastasis is another type of tumor. Please, clarify this while reviewing this section.
Thank you very much.
Author Response
Dear Authors,
This is an interesting review describing the potential therapeutic effects of melatonin in bone associated diseases. In general the review was well-developed, however, I suggest the following changes:
After talking about HSPC, I would include the osteoporosis section. The cancer section should be the last one.
Ans: We thank the Reviewer for this insightful suggestion and we have accordingly changed the order of sections within the review, as per the suggestion.
Respect to the cancer section, you mostly focused on metastasis. I would include more references related to melatonin and bone cancer treatment first (osteosarcoma, Ewing Sarcoma,...), and after this sub-section, developed the bone metastasis sub-section, as the origin of metastasis is another type of tumor. Please, clarify this while reviewing this section.
Thank you very much.
Ans: We thank the Reviewer for this suggestion and we have accordingly revisited the text in the cancer section. The first part of that section now discusses evidence alluding to the potential of melatonin as a treatment for primary bone treatment, followed in the next part by evidence of its effects in metastatic tumors.
Round 2
Reviewer 1 Report
Manuscript is now suitable for acceptance